# Investigation of Best Practices for Venom Toxin Purification in Jellyfish towards Functional Characterisation

**DOI:** 10.3390/toxins15030170

**Published:** 2023-02-21

**Authors:** Blake Lausen, Anahita Ahang, Scott Cummins, Tianfang Wang

**Affiliations:** 1Centre for Bioinnovation, University of the Sunshine Coast, Sippy Downs 4556, Australia; 2School of Science, Technology and Engineering, University of the Sunshine Coast, Sippy Downs 4556, Australia

**Keywords:** venom, medusozoa, jellyfish, toxins, purification, extraction, bioassay, biodiscovery

## Abstract

The relative lack of marine venom pharmaceuticals can be anecdotally attributed to difficulties in working with venomous marine animals, including how to maintain venom bioactivity during extraction and purification. The primary aim of this systematic literature review was to examine the key factors for consideration when extracting and purifying jellyfish venom toxins to maximise their effectiveness in bioassays towards the characterisation of a single toxin.An up-to-date database of 119 peer-reviewed research articles was established for all purified and semi-purified venoms across all jellyfish, including their level of purification, LD50, and the types of experimental toxicity bioassay used (e.g., whole animal and cell lines). We report that, of the toxins successfully purified across all jellyfish, the class Cubozoa (i.e., *Chironex fleckeri* and *Carybdea rastoni*) was most highly represented, followed by Scyphozoa and Hydrozoa. We outline the best practices for maintaining jellyfish venom bioactivity, including strict thermal management, using the “autolysis” extraction method and two-step liquid chromatography purification involving size exclusion chromatography. To date, the box jellyfish *C. fleckeri* has been the most effective jellyfish venom model with the most referenced extraction methods and the most isolated toxins, including CfTX-A/B. In summary, this review can be used as a resource for the efficient extraction, purification, and identification of jellyfish venom toxins.

## 1. Introduction

Animal venom is a proven source of potent bioactive compounds, many of which have been adapted into pharmaceuticals. Specifically, 11 venom-derived drugs are commercially available and approved by both the United States Food and Drug Administration and the European Medicines Agency [1]. The wide range of venom bioactivity can be seen in these different drugs being utilised in the treatment of coronary syndrome, chronic pain treatment, diabetes, hypertension, and sciatica [1]. Only one of these drugs, Prialt^®^, comes from a marine organism (*Conus magus*), which is symptomatic of the widening gap between terrestrial and marine venom research. This deficiency in marine venom research can be anecdotally attributed to the difficulties of working with marine venom in contrast to terrestrial venom. These difficulties include the availability of target organisms in the wild (often seasonal) and poor knowledge of aquaculture of the organisms and extraction procedures from tissue(s) that maintain bioactivity. Regarding the extraction of biomolecules from tissues, a significant diversity of methodologies exists in the literature for jellyfish venom.

Medusozoa represents the clade of organisms within the phylum Cnidaria that contain a dominant free-swimming phase in their life cycle, known as a medusa stage [2,3]. It is this swimming behaviour that gives these organisms the colloquial name of “jellyfish” (also known as sea jellies). There are however several non-jellyfish exceptions within Hydrozoa such as the hydra included within this review, falling under the sub-phylum Medusozoa. These jellyfish have spread across four classes; Cubozoa (box jellyfish), Hydrozoa (hydra and siphonophores), Scyphozoa (true swimming jellyfish), and Staurozoa (stalked jellyfish) [2,3]. Medusozoans deliver their venom using cells called cnidocytes, containing a ballistically discharging organelle called a nematocyst to inject venom into their prey [4]. The cnidocytes capsules are discharged as a response to either a chemical signal or mechanical stimulus that is triggered by contact with prey organisms [5].

Jellyfish venoms are composed of many complex proteins, peptides, and small molecules, each with extensive and sometimes specific bioactivities [4]. This is mainly attributed to the inability for cognitive selection during their predation, requiring their venom to have cytotoxic, cardiotoxic, and neurotoxic effects across a wider range of prey [6]. As such, some jellyfish have venom so benign to humans they are referred to as “non-venomous”, while others such as box jellyfish and Irukandji have venom so powerful, they are well known to cause death in humans within hours [7]. This wide range of bioactivities observed across both the proteinaceous and non-proteinaceous components of jellyfish venom is of emerging focus towards not only biodiscovery but also the development of more effective first aid and clinical treatments, involving the development of an anti-venom [8,9,10,11].

While many toxins have been reported across various different jellyfish species using bottom-up proteomic investigations, only a few have been confirmed as venomous in vitro via purification and subsequently had their native function characterised. Venoms can often exhibit complex structures, such as multiple disulphide bridges and post-translational modifications and, as such, often require a purification from the whole native venom to confirm bioactivity, prior to commitment towards synthetic or recombinant production. Although multiple reviews have comprehensively covered the properties and specific function of known jellyfish venoms [12,13,14,15,16,17,18,19], to date, no systematic review has compared the methods used to extract individual venom, nor compared 50% lethal dose (LD50) of functional assays to identify the best purification method. As such, it is unclear if there are key factors to consider for primary extraction or purification that may ensure the native structure is maintained.

The primary aim of this systematic [20] literature review was to examine the key factors for consideration for the extraction and purification of native venom toxins to maximise their effectiveness in bioassays towards the development of a more standardised methodology. This knowledge could make the biodiscovery of jellyfish venom toxins more appealing to researchers, particularly those unable to justify the financial commitment of peptide synthesis based entirely on either in silico or bottom-up proteomic analysis. In the process, we have developed an up-to-date database of all purified and semi-purified venoms across all venoms within the literature captured (up until 2023), including their LD50, the current status of venom purification, and existing bioassay models.

## 2. Results

### 2.1. Captured Literature

The captured literature database contained 119 peer-reviewed research articles and 5 literature reviews that have reported the purification of toxins of jellyfish supported by bioassays. Of the reviewed literature, the first reported purification of a jellyfish venom and its assay was in 1975 (Figure 1), in which venom from *Stomoiophus meieagris* was fractionated and Na^+^ transporting influencing factors were identified [21]. Relevant literature gradually increased from 2001, peaking in 2012 (13 publications), and then trending downwards up to now. This decrease may be attributed to the inherent difficulties associated with jellyfish venom extraction and purifications relative to terrestrial venoms which is further discussed later in this review. This is best highlighted by the vast majority of the captured articles including a recommendation for the continuation of the research, yet the venom and toxin fractions they identified did not receive any subsequent publications detailing further refinement. While speculative, this demonstrates the difficulty of jellyfish purification in the initial stages of research, regularly causes the delivery of results that do not justify further research into the venom when more favourable sources of bioactive molecules may be present. Overall, the class Scyphozoa represented the most common jellyfish class for venom purifications (90 publications), followed by Cubozoa (39 publications) and Hydrozoa (12 publications). No venom had been reported purified from any Staurozoa.

### 2.2. Jellyfish Venom Identification

A total of 202 venom proteins have been reported from jellyfish, with varying degrees of purification (Table 1), which could be classified into (1) crude venom, (2) a venom fraction, and (3) isolated venom toxins. Crude venom was defined as venom that had been removed from nematocysts, filtered and/or sonicated and/or centrifuged, and intended to be representative of a jellyfish’s entire venom toxin repertoire (i.e., the entire venome). A venom fraction was defined as crude venom that had had its components separated (e.g., liquid chromatography) to refine the potential number of venom toxin products. Finally, isolated toxins were defined as a single homogenous product, pure enough to obtain a single kilodalton (kDa) measurement, which varied from 1 to 600 kDa. For each classification of purification, the venom had additionally been tested in toxicity assays to obtain an LD50 (or LD50 equivalent for cellular assays). Of the 202 reported venoms, 99 (49%) were crude venom, 62 (31%) were venom fractions, and 38 (20%) were isolated venom toxins.

## 3. Discussion

### 3.1. Purification Levels of Jellyfish Venom

A total of 31 jellyfish species had at least one venom purification level reported (Table 1 and Figure 2), with *Chironex fleckeri, Chrysaora quinquecirrha* (Atlantic Sea nettle), and *Cyanea capillata* (Lion’s mane jellyfish) most extensively investigated. *C. quinquecirrha* shares *C. fleckeri’s* status as a ‘medically relevant’ jellyfish [31], due to its rapidly acting cardiotoxic venom [40] and powerful haemolytic properties [36,37]. In addition, it inhabits the East Coast of the United States and the Gulf of Mexico, thereby serving as a convenient model for venom researchers in North America. The observed abundance of *C. capillata* venom research follows a similar logic, with a powerful cardiotoxic venom [40,52,54,57] and a niche distribution within the northern oceans, extending to the Artic, which helps to explain their disproportionately large representation in the literature.

The data highlight that *Aurelia aurita, Cassiopea xamachana, Catostylus mosaicus, Cyanea nozakii, Nemopilema nomurai*, and *Pelagia noctiluca* have had toxic fractions identified, yet not a single toxin purified. These jellyfish represent an excellent opportunity for research as there exists a strong foundation of effective venom purification and fractionation. Moreover, these jellyfish only represent 32 species of over 3500 known jellyfish species from which no, or very little, research has been performed on their venom. Although hydrozoans represent around 77% of all medusozoan species [2,3], they only occupied 12% of the venom purifications reported.

### 3.2. Jellyfish Venom Extraction

Jellyfish venom for subsequent purification of single toxins and toxicity bioassay confirmation have been successfully obtained using a variety of different extraction and purification procedures (Table 2). As categorised by Carrette and Seymour (2004) [33] venom extraction methods can be separated into, natural discharge of intact nematocysts, mechanical disruption of nematocysts, chemically induced discharge of intact nematocysts and removal of liquid from individual intact nematocysts [33]. Only natural and mechanic methods of extraction returned individual toxins.

The most commonly referenced isolation methods [22,23,37,89,94,113,114] were developed by Burnett et al. (1992) [43] and Bloom et al. (1998) [26], the latter of which had been refined from Burnett’s original method. Briefly, jellyfish tentacles were freshly removed from beachside individuals, then immediately refrigerated within two volumes of seawater for 1–4 days. For venom extraction, the tentacle samples were shaken, and an aliquot of water was taken daily for filtration with filter paper prior to being viewed under a microscope to confirm the release of intact cnidocytes from tissue. The cnidocytes were then lyophilised and stored (−70 °C) prior to extraction. The venom was extracted from the cnidocytes via sonication (3 × 20 s with 1 min intervals) in ice-cold water (4 °C) before centrifugation at 20,000× *g* for 1 h at 4 °C and then immediately used in toxicity assays. This approach was regularly referred to as the ‘autolysis’ venom extraction technique since it relied on autolysis of tentacle tissue (but not the cnidocytes themselves). Variations in autolysis venom extraction included tentacle incubation times of one day [88] to up to 6 weeks [82]. However, the longest time of autolysis that yielded a single proteinaceous toxin was 4 days. Therefore, it is recommended that autolysis be no longer than 4 days.

The efficiency of the venom extraction autolysis technique may be attributed to a few factors. First, it has been speculated that the intracellular environment of cnidocytes, including high pressure, contributes to venom stability. In addition, venom is known to act rapidly to facilitate processes that disable prey; therefore, they do not require a high degree of structural stability that could maintain a longer half-life outside of the cellular environment [26].

A variation of the autolysis technique involved the agitation of jellyfish tentacles in a buffered solution for up to 6 weeks to allow for maximal separation of the cnidocytes. Cnidocytes were then chilled in a pressure cell at 12,000 psi for 15 min, followed by a freeze–thaw cycle to help liberate the venom [82]. While this method is a conventional approach for liberating the intracellular components of a cell for small molecule analysis, it should not be used when isolating proteinaceous toxins from jellyfish venom, as multiple studies have reported that freeze–thaw cycles cause a significant decrease in bioactivity [65,83] particularly for proteinaceous compounds. This is relatively well known, as the phase change of an aqueous environment can cause alterations at the tertiary and quaternary levels of proteins by mechanical disruption of the structure during freezing.

A strict thermal regulation regime appears to play a critical role in the preservation of jellyfish venom bioactivity and, therefore, its potency. This is irrespective of cnidocyte separation since a large number of jellyfish toxins have been successfully isolated using whole tentacle homogenisation [84,91,93,95,98,102] and whole tentacle lysis [83]. Thermal denaturation of jellyfish toxins has been studied to investigate their sensitivity. For example, the activity of the CnPH toxin was significantly reduced when exposed to temperatures above 45 °C [116] and the venom activity of the flame jellyfish (*Rhopilema esculentum*) was found to be reduced at 40 °C [8]. It was additionally inferred during the purification of ClGp1 from lysed cnidocytes that it should be performed as fast as possible to preserve bioactivity [98].

Whole tentacle homogenisation and lysis use mechanical, electrical, or chemical means to break open cells, and in doing so, cause nematocysts to fire, releasing the venom. Thus, whole tentacle extraction undoubtedly adds ambiguity as to the true source of the toxin and introduces background biomolecules to the subsequent purification process. For instance, the enzyme hydralysin was identified from a *Chlorohydra viridissima* tentacle homogenate [91], yet it is a cytolytic/neurotoxic proteinaceous toxin that originates from a non-cnidocyte source. This, as well as other examples, indicates the possibility that not all jellyfish “venom” is delivered from nematocysts [15,91]. Whole tentacle extraction may be a better choice if the sample size is low as it will provide a more concentrated venom extract. However, if the jellyfish venom is of low toxicity and if the venom readily loses bioactivity under extraction conditions, it may be significantly impacted during the hours or days that the tentacles are suspended in the separation solution.

### 3.3. Jellyfish Venom Purification

A total of 38 individual jellyfish venom toxins have been identified (Table 2). In all of the isolated toxins within the literature-captured jellyfish venom purification studies, liquid chromatography was the primary method of choice, while most of them utilised a combination of at least two chromatography techniques, including HPLC [93], RP-HPLC [6], SEC [7], CEX [94], and AEX [102]. Only two studies required additional chromatography to ensure the purity of a single toxin [7,59]. Non-chromatography purification processes have also been used in tandem with liquid chromatography, using acidic precipitation [102] and a “salting out” protein precipitation method [103]. While these methods were shown to have produced single toxins from crude venom, they should only be used as a final purification step for identification, when the activity and purity of the toxin have been confirmed. This is due to protein precipitation causing the denaturation of proteins, thereby removing any chance to confirm its native function in bioassays.

Size exclusion chromatography was used in more than 70% of the jellyfish toxin purification studies, thereby representing the most common liquid chromatography method. SEC is particularly useful when purifying unknown toxins, as it does not require organic solvents. Organic solvents, such as acetonitrile, methanol, and ethanol, commonly used in RP-HPLC and chloroform in HPLC, are known to modify protein structure, thereby potentially affecting a toxin’s function. Unlike CEX and AEX, SEC does not require prior knowledge of the proteins’ solubility, charge, or isoelectric point and will provide better resolution across a sample with many compounds. SEC can be performed as high-throughput [23] or used as a last step to “polish” toxins [98].

### 3.4. Toxicity Assays of Venom towards Proposed Function

The combined data obtained, from jellyfish purification to identification, indicate that overall direct comparisons should not be performed due to the lack of commonality between studies regarding the toxicity assay (Figure 3). For jellyfish venoms, mice have been the most used animal model system for toxicity characterisation (to establish lethal dosage; Figure 3A). This is consistent with studies on other types of venom (derived from marine or terrestrial organisms), whereby functional toxicity assays using mice are considered relatively cheap with accessible ethical requirements, as well thiersimilar biological, physiological, and symptomatic effects to humans. However, as animal ethics for vertebrate experimentation become stricter, invertebrate animal models (e.g., crayfish, flies, moths, and spiders) may be more widely used. In addition, jellyfish predate most commonly on small fish and crustaceans making them the most biologically relevant models for whole animal toxicity testing.

Erythrocytes are excellent for cell-based toxicity assays to establish haemolytic activity. To assess jellyfish venom toxicity, sheep erythrocytes have been the most used, likely due to their ethical sourcing, relatively low cost, and high availability (Figure 3B). Chicken erythrocytes have also been commonly used, which although non-mammalian, are still nucleated cells that have demonstrated clear comparative results against different types of jellyfish venom [8,40,45,78,88]. Human erythrocytes have also been well used in cell-based toxicity assays, primarily with relevance to investigating cardiovascular and haemolytic impacts on humans; they are also more sensitive to venom compared to sheep erythrocytes [92]. The haemolytic effects have been studied in human erythrocytes to reveal venom protein interactions with membrane lipids, which were proposed to be directly and most likely involved in pore formation [34,71,92,109]. Research investigating these impacts has used various venom purification methods as this model gives autonomy in method designation, although the HU50s (one haemolytic unit, the amount of protein sample required to induce 50% haemolysis, referred to as LD50 in this review) were quite different between methods conducted on the same species [34]. However, the most dependable method was presented by Yanagihara and Shohet (2012) [34], where human erythrocytes were recognised as being a reproducible source for experiments. They showed specific membrane breakdown when interacting with venom toxins; although considering that different jellyfish venoms contain different repertoires of toxins, they consequently have different haemolytic activity that might not be possible to compare; in addition, reportedly, careful consideration should be taken with the enzymes used in bioassays to avoid inhibitory effects on venom proteins [91].

Human cancer cell lines are often used to elucidate the harmful effects of venoms on humans and to understand the molecular mechanisms of venom toxicity, and they have been used on numerous occasions for jellyfish venom toxins [57,65,72,118,123,124,129]. They provide a reliable, high-throughput, convenient, and relatively cheap approach [124], which is primarily relevant for medical and pharmaceutical applications, specifically when it comes to first aid treatments regarding understudied jellyfish venom stings [123,124]. This model, however, has its own limitation, with a bias for the most accessible cell lines available, thus causing variation in the obtained results [57,65,72,118,123,124,129]. The most common mechanistic function reported using these cell lines involved reactive oxygen species (ROS) and cell apoptosis.

However, it was revealed that comparisons using the same animal model and the crude venom of the same species may not aid in elucidating the most effective purification or extraction technique. Separate investigations of *Chironex fleckeri* “crude venom”, described in the articles as crude tentacle venom [28], stock venom [9], nematocyst venom [26], and *Chironex* venom [32], all used near identical methodologies while producing significantly different LD50s in mice of 11, 12, 23.4, and 150 µg/mL, respectively. This variation in LD50 is most likely due to individual variations in venom potency between jellyfish or venom degradation that may have occurred prior to individual capture and isolation. As jellyfish are difficult to capture while in the ocean, especially smaller species, specimens are most often collected after they have washed up on the beach, where animal death and thermal desiccation quickly degrade the proteinaceous compounds within the venom. Furthermore, as very few jellyfish species are able to be maintained in captivity, repeated bioassays using the same individual animal’s venom is not possible, thus compounding the source of error due to individual variability of venom potency. Although not captured in the scope of this review, it is important to highlight a 2019 study of *Nemopilema nomurai* which showed that individual nomurai venom showed significantly different enzymatic metalloproteinase activity, with up to a 77-fold difference in haemolytic activity [130]. Moreover, the geographical location and the jellyfish’s age are correlated with significant variations in jellyfish venom of the same species [131].

Published studies have noted difficulties associated with the attempted analysis of reported jellyfish venom due to accidental misreporting of the specific activity of venom and the vastly different methodologies involved in the preparation and bioassays [82]. Complicating analysis further, jellyfish venom can have a unique haemolytic activity profile that can be highly variable between jellyfish species [65,116] and that very rarely aligns with general cytotoxicity assay results. Furthermore, the objective of many research studies was not to classify the general function or toxicity of the venom towards the discovery of a toxin. Instead, their objective was focused on the biodiscovery of a compound with a predetermined, specific function: for example, studies of jellyfish venom targeting angiotensin I, converting enzyme (ACE) inhibitors [76,93,112]. As such, jellyfish venom studies currently lack a standard methodology for toxicity measurements in cells and animal models using LD50 measurements. Function-specific bioassays of toxins are too specific and lack any type of standardisation, meaning that the comparison of bioassay data between articles for the purpose of elucidating the most effective purification techniques may not be possible. This highlights that jellyfish venom LD50s should not be used as a means to compare the effectiveness of venom purification or extraction methods. The limitations of venom potency can be best mitigated by obtaining the largest possible sample size when collecting venom for extraction and excluding juvenile or desiccated jellyfish from extractions.

### 3.5. Chironex fleckeri, a Model for Jellyfish Venom Purification

The box jellyfish, *Chironex fleckeri*, is often referred to as the most medically relevant jellyfish [17,31,111], given that its powerful haemolytic and cardiotoxic toxins have, to date, been prime sources for biodiscovery. In fact, the increase in jellyfish venom purification studies can generally be attributed to the attention brought to the field by extensive research performed on cardiotoxic venoms derived from *C. fleckeri* (Figure 2). The box jellyfish is well known for an extremely lethal and fast-acting venom that can be lethal to humans and prey alike. Thus, *C. fleckeri* represents the most researched jellyfish, with the most venomous LD50 score, and from which the most jellyfish toxins have been identified. Although there are outcomes that have reported on the toxin properties and lethality of the whole venom [9,10,23,24,27,28,29,30,32,33,34,99] leading to many bioactive fractions [9,24,25,26,27], the primary lethal toxin responsible for human fatalities has not been reported. Nonetheless, *C. fleckeri* represents a well-established model for the purification of jellyfish toxins and could be further used by researchers for methods of venom characterisation of Medusozoan.

## 4. Conclusions

This review has systematically compiled all kinds of Medusozoan purified venoms (crude venom, toxic fractions, and single toxins), which may serve as a valuable resource for venom researchers. Despite initial intensive foundational research since 1974, with research peaking in 2011, very few toxins have been successfully purified across the Medusozoans, with the exceptions being *C. fleckeri* and *C. rastoni*. Importantly, in order to characterise a single toxin when guided by bioassays, a strict thermal management plan is required to avoid loss of potency, thereby improving the success rate of purifications. Although a clear universal methodology for venom purification could not be elucidated, there is a clear trend for two-step liquid chromatography involving SEC to isolate a single toxin. This variation in the methodologies used, as well as a lack of duplication and the natural variation in individual species’ venom, currently precludes our ability to compare bioassay data. This points towards jellyfish venom extraction and purification methods being entirely dependent upon the observed chemical and functional behaviour of the toxins of interest. Therefore, they should be experimentally determined per species via functional bioassays of the toxin, to confirm the conservation of the bioactivity at each step. Performing comparisons of venom LD50, even when using the same model organism and same jellyfish species, has been shown to be of questionable value given the discussed limitations. To date, *C. fleckeri* has been the most effective jellyfish venom model, from which researchers have developed widely used extraction and purification methods, leading to a large number of venoms being categorised. While jellyfish venom research continues to trail behind terrestrial venom research, this does provide ample opportunity for novel discovery.

## 5. Method

Only literature captured using the methodology described herein is referenced in this review to maintain the integrity of the analysis, unless specifically stated. The inclusion criteria were as follows: Scopus, Web of Science, and Pub Med databases were searched using the keywords “purification or purified” and “venom” in “all fields” using the access provided by the University of the Sunshine Coast. Additionally, five secondary search terms, including “Jellyfish”, “Cubozoa” or “Cubozoan”, “Hydrozoa” or “Hydrozoan”, “Scyphozoa” or “Scyphozoan”, and “Staurozoa” or “Staurozoan”, were added and combined to ensure the capture of all jellyfish purification publications for analysis (Figure 4), with duplicate articles removed. The exclusion criteria were applied as follows: (1) if no venom purification was detailed in the publication, (2) if the venom detailed in the publication is not a single Medusozoan venom, and (3) if the venom purification detailed in the publication was non-guided with no bioassays. Using the above methodology, the literature from between 1 January 1975–15 December 2022 was captured as the earliest and latest articles. This methodology followed the Transparent Reporting of Systematic Reviews and Meta-Analysis (PRISMA) [20]; the database of all of the literature information is provided in Appendix A.

## Figures and Tables

**Figure 1 toxins-15-00170-f001:**
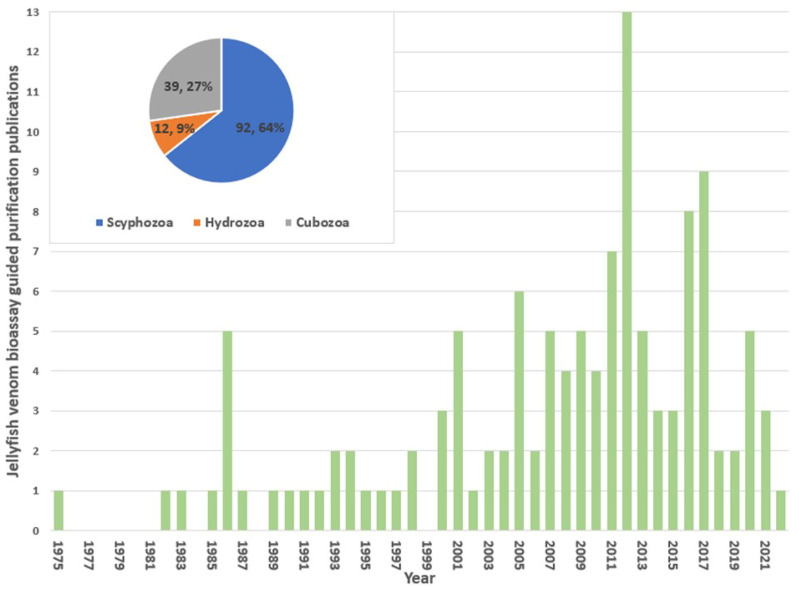
The bar graph shows the number of jellyfish venom purification papers identified per year (1975–2022). In the inset, the pie chart shows the relative distribution (number and percentage) represented by jellyfish class.

**Figure 2 toxins-15-00170-f002:**
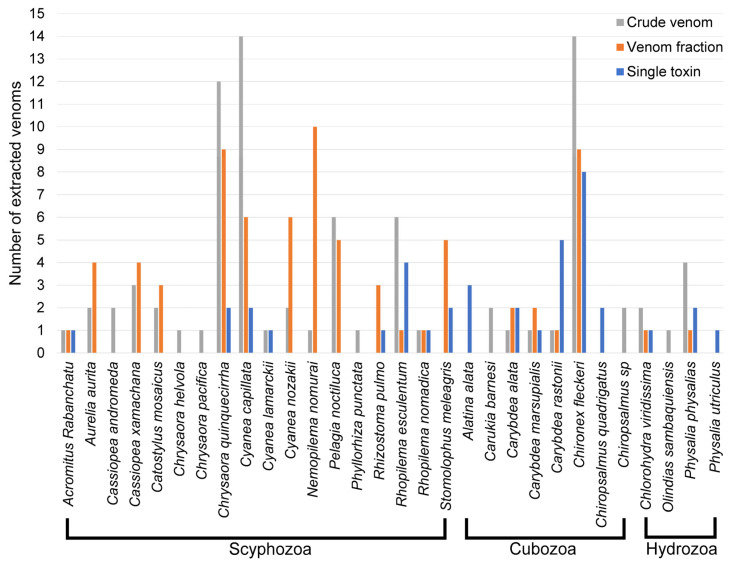
Graph showing a summary of jellyfish species by class (in alphabetical order) and level of their venom purification.

**Figure 3 toxins-15-00170-f003:**
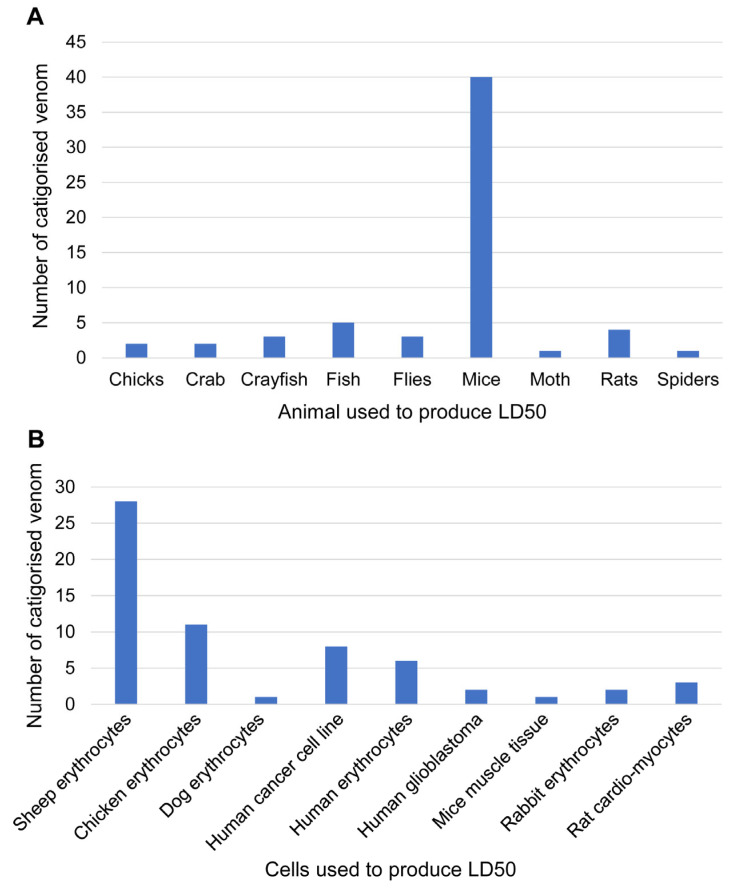
Summary of animal and cell toxicity assays used for bioassay-guided identification of jellyfish venom, including LD50. (**A**) Graph showing the number of venom toxicity assays using whole animal models. (**B**) Graph showing the number of venom toxicity assays using cell-based toxicity assays.

**Figure 4 toxins-15-00170-f004:**
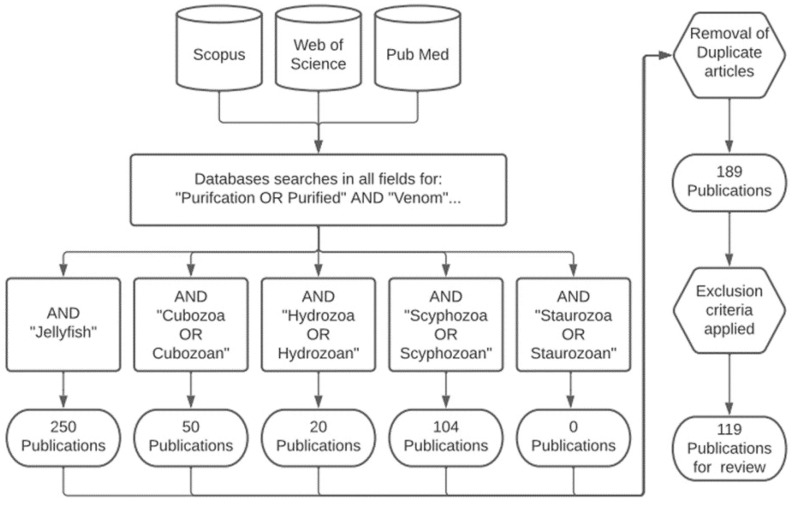
Flowcharts of the systematic method used for this review, producing the 119 publications for analysis.

**Table 1 toxins-15-00170-t001:** Summary of venom or venom toxins isolated from jellyfish, including name, size (kDa), purification level, LD50, and/or reported primary activity. Toxin names have been left abbreviated for clear reference to the original articles. * Non-jellyfish species (hydra).

Species (Class)	Toxin Name and Reference	kDa	Purification Level	LD50 µg/kg	LD50 Assay Animal	LD50 ng/mL	LD50 Cell Type	Reported Primary Activity
*Chironex fleckeri* Southcott, 1956 (Cubozoa)	CfTX-A [22]	40	Isolated toxin	N/A		5	Sheep erythrocytes	Haemolytic
	CfTX-B [22]	42	Isolated toxin					
	Toxin 1 [7]	150	Isolated toxin			N/A		Myotoxic
	Toxin 2 [7]	600	Isolated toxin					
	Major cytolysin 1 [23]	370	Isolated toxin			24	Sheep erythrocytes	Haemolytic
	Major cytolysin 2 [23]	145	Isolated toxin			7		
	Minor cytolysin [23]	70	Isolated toxin			N/A		
	Fraction 2 [24]	N/A	Venom fraction					Cardiotoxic
	T_4_ [25]		Venom fraction					Neurotoxic
	T_5_ [25]		Venom fraction					Haemolytic
	Fraction 11–14 [26]		Venom fraction	3.6	Mice			Unidentified
	Cf Fraction 4 [27]		Venom fraction	N/A				Cytotoxic
	Fraction 11 [26]		Venom fraction	6.5	Mice			
	Flow though fraction [9]		Venom fraction	27		2000	Sheep erythrocytes	Haemolytic
	Specific eluate [9]		Venom fraction	2.2		1100		
	Recovered Fraction [28]		Venom fraction	28		N/A		Cytotoxic
	Whole venom [24]		Crude venom	N/A				Cardiotoxic
	Crude venom [23]		Crude venom			5	Sheep erythrocytes	Haemolytic
	Cf FTNV [27]		Crude venom			N/A		Cytotoxic
	*C. fleckeri* venom [29]		Crude venom	2.2	Fish			Cardiotoxic
	*Chironex fleckeri* crude tentacle extract [30]		Crude venom	N/A				
	*Chironex fleckeri* venom [31]		Crude venom			244 2700	Rat cardiomyocytes Sheep erythrocytes	Haemolytic
	Nematocyst venom [26]		Crude venom	23.4	Mice	N/A		Cytotoxic
	venom [10]		Crude venom	N/A				Neurotoxic
	Chironex venom [32]		Crude venom	150	Mice			Cytotoxic
	Crude tentacle venom [28]		Crude venom	11		670	Sheep erythrocytes	Haemolytic
	Chironex fleckeri [33]		Crude venom	N/A		N/A		Cardiotoxic
	Chironex fleckeri venom [34]		Crude venom	N/A		10.85	Human erythrocytes	Haemolytic
	Stock venom [9]		Crude venom	12	Mice	N/A		Cytotoxic
	*C.fleckeri* venom [35]		Crude venom	N/A				Cardiotoxic
*Chrysaora* *Quinquecirrha* Desor, 1848 (Scyphozoa)	Sea Nettle toxin [36]	105	Isolated toxin	23	Mice			Unidentified
	Chrysaora hemolysin [37]	6–10	Isolated toxin	N/A				Haemolytic
	Cq Fraction 4 [27]	N/A	Venom fraction					Cytotoxic
	Frc-1 [38]		Venom fraction					Antioxidant
	Frc-2 [38]		Venom fraction					
	Frc-3 [38]		Venom fraction					
	toxin fraction [37]		Venom fraction					Haemolytic
	Partially purified Hyaluronidase [39]		Venom fraction					
	HPLC fraction [37]		Venom fraction					
	C. quinquecirrha specific fraction [40]		Venom fraction			1000	Chicken erythrocytes	Cardiotoxic
	Lethal proteins [41]		Venom fraction			N/A		Unidentified
	Cq FTNV [27]		Crude venom					Cytotoxic
	Crude venom [36]		Crude venom	58	Mice			Unidentified
	Crude [38]		Crude venom	N/A				Antioxidant
	Crude cell free extract [37]		Crude venom					Haemolytic
	C. quinquecirrha crude extract [40]		Crude venom			25,000	Chicken cardiocytes	Cardiotoxic
	Sea nettle crude toxin [42]		Crude venom			N/A		Allergen
	Chrysaora quinquecirrha [31]		Crude venom			407	Rat cardiocytes and cardiomyocytes	Cytotoxic
	Liquid N_2_ *Chrysaora quinquecirrha* venom [43]		Crude venom	690	Mice	N/A		
	Chrysaora venom [32]		Crude venom	210				
	SN crude [44]		Crude venom	20	Fish			
	Chrysaora quinquecirrha [33]		Crude venom	Fraction N/A				Cardiotoxic
*Cyanea capillata* Linnaeus, 1758 (Scyphozoa)	CcNT [6]	8.22	Isolated toxin					Neurotoxic
	Major fraction [45]	N/A	Venom fraction			13,500 8800	Chicken erythrocytes and rabbit erythrocytes	Haemolytic
	toxin fraction [37]		Venom fraction			N/A		
	HPLC fraction [37]		Venom fraction					
	C. capillata specific fraction [40]		Venom fraction			9000	Chicken erythrocytes	Cardiotoxic
	Fraction 3 [45]		Venom fraction			N/A		Neurotoxic
	Fraction F3B [46]		Venom fraction					
	C. capillata crude extract [40]		Crude venom			6000	Chicken cardiocytes	Cardiotoxic
	TE [47]		Crude venom			N/A		Phosphorylation
	TE [48]		Crude venom					Cytotoxic
	TOE [49]		Crude venom			156,000	Sheep erythrocytes Erythrocytes	Haemolytic
	TE [50]		Crude venom			N/A		Cytotoxic
	TE [51]		Crude venom					
	TOE [52]		Crude venom					Cardiotoxic
	TOE [53]		Crude venom	4250	Mice			Haemolytic
	C TOE [54]		Crude venom	N/A				Cardiotoxic
	TOE [55]		Crude venom					Haemolytic
	TE [56]		Crude venom					
	CnV [57]		Crude venom			950	Human cancer cell line	Cytotoxic
	C. capillata venom [58]		Crude venom			N/A		
	Crude cell-free extract [37]		Crude venom					Haemolytic
	CcTX-1 [59]		Crude venom					
*Nemopilema* *Nomurai* Kishinouye, 1922 (Scyphozoa)	NnLF [60]		Venom fraction					
	NnV [61]		Crude venom					Proteolytic
	Nemopilema nomurai nematocysts venom [62]		Crude venom			63,620	Sheep erythrocytes	Haemolytic
	NnV [57]		Crude venom			280	Human cancer cell line	Cytotoxic
	NnNV [4]		Crude venom			N/A		Cytolytic
	NnFV [63]		Crude venom	29.1	Spiders			Insecticidal
	NnV [64]		Crude venom	N/A				Cardiotoxic
	N. nomurai venom [65]		Crude venom			2000, 1200, 151,000	Human cancer cell line and dog erythrocytes	Cytotoxic
	NnV [66]		Crude venom			N/A		Cardiotoxic
	NnV [67]		Crude venom					Proteolytic
	N.nomurai venom [68]		Crude venom					
	NnTXs [11]		Crude venom					Haemolytic
*Pelagia noctiluca* Forsskål, 1775 (Scyphozoa)	P. noctiluca specific fraction [40]		Venom fraction			4000	Chicken erythrocytes	Cardiotoxic
	F1 [69]		Venom fraction			125,000	Human glioblastoma U87	Cytotoxic
	F3 [69]		Venom fraction			179,000	Human glioblastoma U87	
	Fraction 1 [70]		Venom fraction			N/A		Analgesic
	Fraction 2 [70]		Venom fraction					
	Fraction II [71]		Venom fraction					Neurotoxic
	*P. noctiluca* crude extract [40]		Crude venom			14,000	Chicken cardiocytes	Cardiotoxic
	Crude venom [70]		Crude venom	20,000	Mice	N/A		Analgesic
	Crude extract [71]		Crude venom	N/A		980	Human erythrocytes erythrocytes	Haemolytic
	Crude venom [72]		Crude venom			300,000	Human cancer cell line	Cytotoxic
	*Pelagia Noctiluca* crude venom [73]		Crude venom			N/A		Nematocysts inhibition
	Crude venom [74]		Crude venom					Haemolytic
	Pelagia noctiluca Crude venom [75]		Crude venom					Haemolytic
*Rhopilema* *Esculentum* Kishinouye, 1891 (Scyphozoa)	RPH [76]		Venom fraction					ACE inhibitor
	Rhopilema esculentum Venom [77]		Crude venom					Cytotoxic
	RNV [78]		Crude venom			910	Chicken erythrocytes	Haemolytic
	RFV [8]		Crude venom			3400		
	CT [79]		Crude venom			12,400	Sheep erythrocytes	
	RFV [80]		Crude venom			N/A		
	R.esculentum full proteinous venom [81]		Crude venom	123.1	Moth			Insecticidal
*Chironex* *Yamaguchii* Lewis and Bentlage, 2009 (Cubozoa)	4A [82]	0.5	Isolated toxin	N/A				No known function
	4B [82]	0.5	Isolated toxin					
	4C [82]	0.5	Isolated toxin					
*Carybdea rastoni* (Cubozoa) Haacke, 1886	CrTX-A [83]	43	Isolated toxin	20	Crayfish	1.9	Sheep erythrocytes	Haemolytic
	CrTX-B [83]	46	Isolated toxin	N/A	N/A	2.2		
	CrTX-I [84]	49	Isolated toxin	3.5	Mice	N/A		
	CrTX-II [84]	100	Isolated toxin	3.6				
	CrTX-III [84]	51	Isolated toxin	3.0				
	pCrTX [84,85]	N/A	Crude venom	127				
*Carybdea* *Marsupialis* Linnaeus, 1758 (Cubozoa)	CARTOX [86]	102	Isolated toxin	N/A		50	Sheep erythrocytes	
*Stomolophus* *Meleagris* Agassiz, 1860 (Scyphozoa)	SmP90 [87]	90	Isolated toxin			N/A		Antioxidant
	SmTX [88]	45–52	Isolated toxin			70,000	Chicken erythrocytes	Haemolytic
	Fraction A [21]	N/A	Venom fraction	>40,000	Mice	N/A		Cardiotoxic
	Fraction B [21]		Venom fraction	>40,000				
	Fraction C [21]		Venom fraction	>40,000				
	Fraction D [21]		Venom fraction	>40,000				
	Fraction E [21]		Venom fraction	32,000				
*Carybdea alata* Reynaud, 1830 (Cubozoa)	CaTX-A [89]	43	Isolated toxin	25	Crayfish	70	Sheep erythrocytes	Haemolytic
	CaTX-B [89]	45	Isolated toxin	N/A		80		
	First peak [90]	N/A	Venom fraction			20		
	Second peak [90]		Venom fraction			25		
	Crude venom [90]		Crude venom			290		
*Rhopilema nomadica* Spanier and Ferguson, 1990 (Scyphozoa)	PhA2 [91]	>10	Isolated toxin	6000	Fish	N/A		Cytolytic
	Crude toxin peak 1 [92]	N/A	Venom fraction	N/A		1250	Human erythrocytes	Haemolytic
	Crude toxin [92]		Crude venom			1250		
*Rhopilema esculentum* Kishinouye, 1891 (Scyphozoa)	X1 [93]	0.434	Isolated toxin			N/A		ACE inhibitor
	X2 [93]	0.683	Isolated toxin					
	X3 [93]	0.754	Isolated toxin					
	X4 [93]	0.778	Isolated toxin					
*Physalia utriculus* Gmelin, 1788 (Hydrozoa)	PuTx-IVC [94]	N/A	Isolated toxin	1190	Mice			Cytolytic
*Physalia physalis* (Linnaeus, 1758) (Hydrozoa)	PpV9.4 [95]	0.550	Isolated toxin	N/A				Promote insulin secretion
	PpV19.3 [95]	4.720	Isolated toxin					
	*P. physalis* specific fraction [40]	N/A	Venom fraction			5000	Chicken erythrocytes	Cardiotoxic
	*P. physalis* crude extract [40]		Crude venom			75,000		Cardiotoxic
	Man-o-war crude toxin [42]		Crude venom			N/A		Allergen
	Portuguese Man-O-War [96]		Crude venom					Cytotoxic
	Nematocyst fluid [97]		Crude venom					
*Cyanea lamarckii* Péron and Lesueur, 1810 (Scyphozoa)	ClGp1 [98]	25.7	Isolated toxin					
	*C. lamarckii* [58]	N/A	Crude venom					
*Aurelia aurita* Linnaeus, 1758 (Scyphozoa)	Fraction A-B [99]		Venom fraction	2000–3000	Mice			Haemolytic
	Fraction 4 [100]		Venom fraction	N/A				Neurotoxic
	Fraction I [101]		Venom fraction					Fibrinogenic
	Major fraction [45]		Venom fraction			35,300 43,100	Chicken erythrocytes and rabbit erythrocytes	Haemolytic
	Crude venom [99]		Crude venom	3200–4200	Mice	N/A		Haemolytic
	JFTE [101]		Crude venom	N/A				Proteolytic
*Acromitus* *Rabanchatu* Annandale, 1915 (Scyphozoa)	T-Ar [102]	182	Isolated toxin	850	Mice			Myotoxic
	Fr-II [102]	N/A	Venom fraction	3000 7700				
	Crude venom [102]		Crude venom					
*Rhizostoma pulmo* Macri, 1778 (Scyphozoa)	Rhizoprotease [103]	95	Isolated toxin	N/A				Proteolytic
*Carybeda rastonii* Haacke, 1886 (Cubozoa)	pCrTX [104]	N/A	Venom fraction	127	Mice			Cardiotoxic
*Chlorohydra* *Viridissima* * Pallas, 1766 (Hydrozoa)	Pure Toxin [105]	27	Isolated toxin	3.6	Flies			Cytolytic
	first separation [105]	N/A	Venom fraction	156				
	BE [105]		Crude venom	75.2				
*Carybdea marsupialis* Linnaeus, 1758 (Cubozoa)	Fraction A [106]		Venom fraction	N/A				Neurotoxic
	CmNt [5]		Venom fraction	15	Crab			
	Crude extract [5]		Crude venom	1050				
*Cassiopea xamachana* Bigelow, 1892 (Scyphozoa)	Fraction III [107]		Venom fraction	280	Mice			Haemolytic
	Fraction IV [107]		Venom fraction	250				
	Fraction VI [107]		Venom fraction	120				Cytolytic
	C. xamachana post-FACS [108]		Venom fraction	340		7000	Sheep erythrocytes	Haemolytic
	CxTX [107]		Crude venom	750		N/A		Cytolytic
	Crude Cx Venom [109]		Crude venom	N/A		6890 56,000	Human erythrocytes and sheep erythrocytes	Haemolytic
	C. xamachana pre-FACS [108]		Crude venom	1600	Mice	110,000	Sheep erythrocytes	
*Catostylus Mosaicus* Quoy and Gaimard, 1824 (Scyphozoa)	Fraction bound (2) [110]		Venom fraction	N/A		N/A		
	Fraction bound (3) [110]		Venom fraction					
	Fraction bound (4) [110]		Venom fraction					
	CE [110]		Crude venom					
	Blubber venom [111]		Crude venom	2184	Mice			
*Chiropsalmus* *Quadrigatus* Haeckel, 1880) (Cubozoa)	Fraction 3.5 peptide [112]	0.97	Isolated toxin	>2,000,000	Rats	N/A		ACE inhibitor
	CqTX-A [113,114]	44	Isolated toxin	80	Crayfish	160	Sheep erythrocytes	Haemolytic
*Cyanea nozakii* Kishinouye, 1891 (Scyphozoa)	CnLF [115]	N/A	Venom fraction	N/A		N/A		Cytolytic
	CnPH [116]		Venom fraction			5000	Sheep erythrocytes	Haemolytic
	Nematocyst content [117]		Crude venom	600	Fish	N/A	N/A	Neurotoxic
	CnV [115]		Crude venom	316,000	mice	N/A	N/A	Cardiotoxic
	Cyanea nozakii nematocyst venom [62]		Crude venom	N/A		69,690	Sheep erythrocytes	Haemolytic
	CNN [118]		Crude venom			5.1 17.9 24.3	Human cancer cell line	Cytotoxic
	CnNV [4]		Crude venom			N/A		Cytolytic
	C.nozakii venom [68]		Crude venom					Proteolytic
*Rhizostoma pulmo* Macri, 1778 (Scyphozoa)	SP [119]		Venom fraction					Cytotoxic
	SP > 30 [119]		Venom fraction					
	Fraction I [103]		Venom fraction					Fibrinogenic
*Carukia barnesi* Southcott, 1967 (Cubozoa)	CVE [120]		Crude venom	N/A				Cardiotoxic
	C. barnesi venom [35]		Crude venom					Cytotoxic
*Cassiopea andromeda* Forskål, 1775 (Scyphozoa)	Cassiopea andromeda venom [121]		Crude venom	104.0	Mice			Haemolytic
	Crude tentacle-only extract [122]		Crude venom	104.0	Rats			
*Chrysaora helvola* Brandt, 1838 (Scyphozoa)	NV [123,124]		Crude venom	N/A		3130, 220	Human cancer cell line	Cytotoxic
*Chrysaora pacifica* Goette, 1886 (Scyphozoa)	CpV [125]		Crude venom			N/A		Neurotoxic
*Phyllorhiza* *punctata* Lendenfeld, 1884(Scyphozoa)	Crude Protein Extract [126]		Crude venom					
*Olindias* *Sambaquiensis* Müller, 1861 (Hydrozoa)	Crude venom [127]		Crude venom					Proteolytic
*Chiropsalmus* sp. Agassiz, 1862 (Cubozoa)	*Chiropsalmus* sp. venom [29]		Crude venom	60.370	Fish			Cardiotoxic
	*Chiropsalmus* sp. venom [10]		Crude venom	N/A				Neurotoxic
*Chrysaora* sp. Desor, 1848 (Scyphozoa)	*Chrysaora* sp. venom [128]		Crude venom					Haemolytic
*Phacellophora camtschatica* Brandt, 1835 (Scyphozoa)	Tentacle extract [129]		Crude venom	3290	Mice	400,290 92,440	Mouse erythrocytes and human cancer cell line	Haemolytic

**Table 2 toxins-15-00170-t002:** Summary of 38 toxins successfully isolated, including extraction and purification procedures. SEC, size-exclusion chromatography; CEX, cation exchange chromatography; AEX anion exchange chromatography; HPLC, high-performance liquid chromatography; RP-HPLC, reverse phase high performance liquid chromatography.

Species (Class)	Toxin Name	Separation Solution	Cnidocyte Separation	Venom Extraction	Purification Step 1	Purification Step 2
*Chionex fleckeri* (Cubozoa)	CfTX-A [22]	Seawater	4-day autolysis (Bloom)	Sonication 20s x3 on ice	SEC	CEX
	CfTX-B [22]					
*Alatina alata* (Cubozoa)	4A [82]	1M NaCl	3–6 weeks autolysis	Omotic pressure using pressure cell		RP HPLC
	4B [82]					
	4C [82]					
*Chiropsalmus quadrigatus* (Cubozoa)	CqTX-A [113,114]	Seawater	4-day autolysis (Burrnet)	Sonication in MQ	CEX	CEX
*Carybdea rastoni* (Cubozoa)	CrTX-A [83]	Whole tentacle Osmotic lysis		Sonication in 5 mM phosphate buffer		
	CrTX-B [83]					
*Carybdea marsupialis* (Cubozoa)	CARTOX [86]	Distilled water	Osmotic pressure 5 min	Sonication in MQ x6		
*Carybdea alata* (Cubozoa)	CaTX-A [89]	Seawater	4-day autolysis (Burrnet)	Sonication in MQ on ice		
	CaTX-B [89]					
*Stomolophus meleagris* (Scyphozoa)	SmP90 [87]		1-day autolysis	Sonication in extraction buffer	AEX	SEC
*Rhopilema nomadica* (Scyphozoa)	PhA2 [91]	Tentacle homogenisation		Omotic pressure dialysis		CEX
*Rhopilema esculentum* (Scyphozoa)	X1 [93]			Omotic pressure MQ	HPLC	SEC
	X2 [93]					
	X3 [93]					
	X4 [93]					
*Physalia utriculus* (Hydrozoa)	PuTx-IVC [94]	Seawater	4-day autolysis (Burrnet)	Sonication in MQ on ice	CEX	CEX
*Physalia physalis* (Hydrozoa)	PpV9.4 [95]	Tentacle homogenisation		Omotic pressure MQ	SEC	RP HPLC
	PpV19.3 [95]					
*Cyanea lamarckii* (Scyphozoa)	ClGp1 [98]			Sonication acetate	Lectin-affinity chromatography	SEC
*Chironex fleckeri* (Cubozoa)	Toxin 1 [7]	Unknown	Unknown	Grinding	SEC	SECx5
	Toxin 2 [7]					
*Acromitus Rabanchatu* (Scyphozoa)	T-Ar [102]	Tentacle homogenisation		Freeze–thaw	Acidic precipitation	AEX
*Cyanea capillata* (Scyphozoa)	CcNT [6]	Distilled water	10 h	Sonication in extraction buffer	SEC	RP HPLC
*Chironex fleckeri* (Cubozoa)	Major cytolysin 1 [23]	Seawater	4-day autolysis (Bloom)	Sonication 20s x3 on ice		CEX
	Major cytolysin 2 [23]					
	Minor cytolysin [23]					CEX
*Chiropsalmus quadrigatus* (Cubozoa)	Fraction 3.5 peptide [112]	Distilled water	2 days autolysis			RP HPLC
*Chrysaora quinquecirrha* (Scyphozoa)	Sea Nettle toxin [36]	1.5% NaCl	4-day autolysis	Grinding		SEC
	Chrysaora hemolysin [37]	Seawater	4-day autolysis (Burrnet)	Sonication in MQ on ice		N/A
*Cyanea capillata* (Scyphozoa)	CcTX-1 [59]	Distilled water	10 h	Sonication in extraction buffer		CEX RP HPLC
*Rhizostoma pulmo* (Scyphozoa)	Rhizoprotease [103]	Tentacle homogenisation		10H Autolysis	Salting-out precipitation	SEC
*Stomolophus meleagris* (Scyphozoa)	SmTX [88]	Distilled water	1 day	Sonication in extraction buffer	AEX	SEC
*Carybdea rastoni* (Cubozoa)	CrTX-I [84]	Tentacle homogenisation			HPLC	
	CrTX-II [84]					
	CrTX-III [84]					
*Chlorohydra Viridissima* (Hydrozoa)	Hydralysin [105]			Grinding	AEX	AEX

## Data Availability

All research data are included within the Appendix A.

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
