# Peer review of "Investigation of Best Practices for Venom Toxin Purification in Jellyfish towards Functional Characterisation"

_toxins, 2023, doi:10.3390/toxins15030170_

Round 1
Reviewer 1 Report
The authors have described a systematic approach and reviewed 119 research articles; providing valuable information about jellyfish venom purification and a model of study for bioassay. Overall, the work is straightforward and easy to follow; however, there are some significant weaknesses of this study that would have to be addressed before this is acceptable for publication.
Point 1
To give the taxonomic authorities the credit that they deserve, it is important to add the taxonomic authorities and the date of description in the manuscript. These can be found by simply typing in the organism’s name into Google and citing the species' name. For example, Chironex fleckeri should be written Chironex fleckeri Southcott, 1956, the first time it is cited, but thereafter not again.
Point 2
Medusozoa is not equivalent to jellyfish. Medusozoa is a subphylum that includes several classes including jellyfish; however, not all medusozoans are jellyfish. Some hydrozoans, such as those species belonging to the genus Millepora (Linnaeus, 1758), are not classified as jellyfish.
Point 3
The use of the word “All” is ambiguous in several sentences.
Line 81 … all purified … Does it mean totally purified toxins or all the venom being studied?
Line 233 – 234 … In all jellyfish venom purification studies … Would it be better to use “all the jellyfish venom purification studies mentioned”?
Line 347 – 348 … all Medusozoan purified venoms (crude venom, toxic fractions, and single toxins)... Would it be better to use “all kinds of Medusozoan purified venoms”?
Point 4
Some of the information is not up to date.
Line 45 to 47 According to the WoRMS information, there are 48 species of Class Cubozoa, instead of 45. Please update the information (as well as those for the other classes).
Reviewer 2 Report
I have carefully examined the manuscript entitled “Investigation of best practices for venom toxin purification in 2 Medusozoans (jellyfish) towards functional characterisation”, submitted as review on Toxins. In this review authors, after a systematic refining, collected 119 articles on Medusozoa in the period 1975-2022 from Scopus, Web of Science, and Pub Med databases, and proposed a novel analytic approach in presenting the data coming from the selected papers. In particular, due to the presence in literature of many reviews reporting properties, function and, in some cases, characterization of jellyfish venoms, authors focused their attention on the key issues in the extraction and purification conditions of native venom toxins and the correlation with their biopharmacological efficacy.
With this method, they have produced a sort of guide for researchers, based on the development of a database of purified and semi-purified jellyfish venoms reporting their LD50, extraction and purification methods, and identification, useful to choose the best approach for an efficient chemical and biological analysis of jellyfish venoms.
Authors performed an accurate comparative analysis among the selected literature, giving as a result an interesting novel view on the Medusozoan venoms.
The manuscript is written in a good and clear English language.
Even if many other reviews on jellyfish toxins are already reported in literature, this work could be considered as an interesting contribution to this field, due to its novel methodological classification.
On this basis, I will consider the above-mentioned manuscript suitable for publication on Toxins.
